# Bacteria in Soil: Promising Bioremediation Agents in Arid and Semi-Arid Environments for Cereal Growth Enhancement

Abdelwahab Rai [1], Mohamed Belkacem [2], Imen Assadi [3], Jean-Claude Bollinger [4], Walid Elfalleh [3], Aymen Amine Assadi [5], Abdeltif Amrane [5,*] and Lotfi Mouni [1]

1  Laboratoire de Gestion et Valorisation des Ressources Naturelles et Assurance Qualité, Faculté SNVST, Université de Bouira, Bouira 10000, Algeria
2  Faculté SNVST, Université Akli Mohand Oulhadj, Bouira 10000, Algeria
3  Laboratoire Energie, Eau, Environnement et Procèdes (LR18ES35), ENIG, Université de Gabès, Gabès 6072, Tunisia
4  Laboratoire E2Lim, Université de Limoges, 123 Avenue Albert Thomas, 87060 Limoges, France
5  Ecole Nationale Supérieure de Chimie de Rennes, University Rennes, CNRS, ISCR-UMR 6226, 35000 Rennes, France
*  Correspondence: abdeltif.amrane@univ-rennes1.fr

**Abstract:** In arid regions, starchy agricultural products such as wheat and rice provide essential carbohydrates, minerals, fibers and vitamins. However, drought, desiccation, high salinity, potentially toxic metals and hydrocarbon accumulation are among the most notable stresses affecting soil quality and cereal production in arid environments. Certain soil bacteria, referred to as Plant Growth-Promoting Rhizobacteria (PGPR), colonize the plant root environment, providing beneficial advantages for both soil and plants. Beyond their ability to improve plant growth under non-stressed conditions, PGPR can establish symbiotic and non-symbiotic interactions with plants growing under stress conditions, participating in soil bioremediation, stress alleviation and plant growth restoration. Moreover, the PGPR ability to fix nitrogen, to solubilize insoluble forms of nutrients and to produce other metabolites such as siderophores, phytohormones, antibiotics and hydrolytic enzymes makes them ecofriendly alternatives to the excessive use of unsuitable and cost-effective chemicals in agriculture. The most remarkable PGPR belong to the genera *Arthrobacter*, *Azospirillum*, *Azotobacter*, *Bacillus*, *Enterobacter*, *Klebsiella*, *Pseudomonas*, etc. Therefore, high cereal production in arid environments can be ensured using PGPR. Herein, the potential role of such bacteria in promoting wheat and rice production under both normal and derelict soils is reviewed and highlighted.

**Keywords:** cereals; induced systemic tolerance; rhizosphere; soil bacteria; pollution





## 1. Introduction

In 2019, the United Nations Organization signaled that demographic growth had reached eight billion inhabitants on the planet, where most of the population live in arid and semi-arid environments (Asia: 60% and Africa: 16%). In addition, the global population is expected to reach 8.5 billion by 2030, 9.7 billion by 2050 and 10.9 billion by 2100 [1]. This rapidly growing population requires increasing food production, essentially coming from agriculture. Thus, a doubling of food and feed production is needed in the next forty years to respond to these new requirements [2].

Arid environments are extremely diverse in terms of landforms, soils, fauna, flora, water balances and human activities. Thus, no practical definition of arid environments can be derived, whereas the one binding element to all arid regions is "aridity". It is important to underline the fact that arid agriculture receives rising attention because of its related problems in mostly undeveloped countries; where a third of all human beings live in 41% of the globe's surface, getting the larger part of their food from cereals and vegetables [3].

The increase in aridity decreases water availability, crop yield and agricultural productivity [4]. Several studies have indicated that increased carbon dioxide concentrations in the

atmosphere lead to global warming [5–7]. Consequently, an increase in aridity is predicted in some model scenarios where drought would persist in some areas of the globe [8–10]. According to Le Houérou [11], there has been an increase of 0.5 °C in global temperature over the past 100 years, which is partially due to excessive urbanization and industrialization. Otherwise, abiotic stresses such as drought, salinity, extreme temperatures, chemical toxicity and oxidative stress are serious threats to agriculture and the natural status of arid environments [12].

In addition, several human activities related to industry, such as the excessive use of petroleum, but also to other agricultural practices such as the hysterical application of fertilizers, pesticides and herbicides, have improved human life quality. However, such activities have also led to the accumulation of alarming amounts of salts and other toxic chemicals, leading to environmental degradation and soil deprivation [13]. These phenomena do not only affect plant growth, but also the soil's microbial community, including beneficial soil bacteria. In this context, the results obtained by Maestre et al. [14] suggest that changes in aridity, following the climate-change models, may reduce microbial abundance and diversity, a response that will likely impact soil fertility and climate regulation. Therefore, several practices have been adopted over time to remediate soils and to enhance plant growth under stress conditions. Among these solutions, certain soil bacteria, referred to as Plant Growth-Promoting Rhizobacteria (PGPR), can colonize the surfaces or inner tissues of plant roots, providing beneficial advantages for both soils and plants [15]. Beyond their ability to improve plant growth under non-stressed conditions, some PGPR are able to establish symbiotic and non-symbiotic interactions with plants growing under stress conditions, participating in soil bioremediation, stress alleviation and plant growth restoration [16]. The ability of PGPR to fix nitrogen, to solubilize nutrients and to produce metabolites such as siderophores, phytohormones, antibiotics and hydrolytic enzymes makes them one of the most ecofriendly alternatives to avoid the excessive use of unsuitable and cost-effective chemicals in agriculture [17]. In the last few years, PGPR have retained both scientists' and farmers' attention as interesting substitutes for chemicals for their sustainable and healthy effect on the environment, but also their promising roles in bioremediation [18,19].

Recently, some reviews have highlighted the role of PGPR as abiotic stress alleviators in soil and the prospects of their application to mitigate soil metal contamination [20–22]. However, none was directed to describe their impact on soil bioremediation and cereal growth enhancement in arid environments. In this paper, we highlight the importance of using beneficial soil bacteria for both soil quality restoration and plant growth enhancement in arid environments. We also summarize scientific works revealing the place of such soil bacteria in improving wheat and rice production under stress circumstances. It is important to mention that cereals, particularly wheat and rice, are known to be the most important crops in the world. Together with maize, they constitute more than 50% of all the calories consumed by human beings over the world [23]. In addition, human diets, especially in developing countries where aridity is dominant, are essentially based on cereals. For example, global cereal demand is expected to increase from 585 million to 828 million tons by 2025, corresponding to an increase of 42% [2]. Furthermore, developing countries' rice and wheat production is supposed to jump from 4.2 and 3.1 to 4.7 and 3.5 tons/ha, between the year 2015 and 2030, respectively. Such rising cereal production would seem to be unsatisfactory to meet the accelerated growth of the human population that is predicted to reach 8.5 billion by the end of 2030 [1,24].

Accordingly, this review aims to understand the prospective functions of Plant Growth-Promoting Rhizobacteria (PGPR), which can participate in soil bioremediation, stress alleviation and plant growth restoration. In fact, the positive effects of PGPR could be exploited to promote agricultural practices in stressed environments and help specialists to manage decisions for more ecofriendly practices in agriculture.

## 2. Main Aspects of Low Soil Fertility

With industrialization, anthropogenic activities such as crude oil exploitation, mining, urban development and the excessive use of chemical fertilizers, pesticides and herbicides have strongly affected soil's physical, physiological and biochemical properties, but also its intrinsic heterogeneous microbial diversity [13,25]. Compounds resulting from such activities are known to be hardly biodegradable and contain high amounts of potentially toxic elements (hereafter: PTE) and other pollutants, hence persisting in nature and affecting vegetal development [19]. In addition, the microbial communities in these contaminated soils are disturbed, which affects their important roles in organic matter recycling, plant disease control, vegetal growth enhancement and the detoxification of the deleterious chemicals in the soil [26,27]. Among PTEs, lead (Pb), chromium (Cr), arsenic (As), zinc (Zn), cadmium (Cd), copper (Cu), mercury (Hg) and nickel (Ni) are the most encountered. Unlike other pollutants, soil is the main tank of such PTE. At high concentrations, these compounds are highly toxic for both plants and microorganisms [28]. Moreover, global warming, together with water scarcity, has led to excessive irrigation and chemical fertilizer application to meet global food requirements, which has resulted in salt accumulation in soil. High concentrations of salt ions in soil, mainly $Na^+$ and $Cl^-$, but also other ions such as ($K^+$, $Ca^{2+}$, $(CO_3)^{2-}$, etc.), reduce water acquisition by plant roots, disturb soil microflora and accentuate phytopathogen virulence [29–31]. Recently, using certain soil bacteria for soil health maintaining and plant growth restauration under abiotic stresses attracts both farmers and scientists as a potential alternative to chemicals. Table 1 summarizes the most recent scientific advances in using beneficial soil bacteria for bioremediation and crop growth restoration.

**Table 1.** Recent advances in soil remediation and plant growth restoration using beneficial soil bacteria.

| Bacterial Genera | Soil-Related Problem | Targeted Crops | Reference |
|---|---|---|---|
| *Bacillus* | Toxic metals (Cr) | *Triticum durum* | Mazhar et al., 2020 [32] |
| *Pseudomonas* *Bacillus* | Toxic metals (As) | *Oryza sativa* | Xiao et al., 2020 [33] |
| *Enterobacter* *Citrobacter* | Toxic metals (Cd, Ni and Pb) | *Triticum aestivum* | Ajmal et al., 2022 [34] |
| *Pseudomonas* *Bacillus* | Toxic metals (Cd, Pb, Zn) | *Spinacea oleracea* L. | Shilev et al., 2020 [35] |
| *Klebsiella* *Pantoea* | Toxic metals (Cd, Zn) | *Pennisetum purpurenum* | Sumranwanich et al., 2022 [36] |
| *Sinorhizobium* *Agrobacterium* | Toxic metals (Cu, Zn) | *Medicago lupulina* | Jian et al., 2019 [37] |
| *Bacillus* *Azotobacter* | Toxic metals | *Pisum sativum* | Singh et al., 2019 [38] |
| *Pseudomonas* *Serratia* | Salt stress | *Triticum durum* | Sohaib et al., 2020 [39] |
| *Kocuria* *Cronobacter* | Salt stress | *Triticum durum* | Afridi et al., 2019 [40] |
| *Pseudomonas* | Salt stress | *Triticum durum* | Albdaiwi et al., 2019 [41] Boumaaza, 2020 [42] |
| *Bacillus* | Salt stress | *Oryza sativa* | Shultana et al., 2021 [43] |
| *Bacillus* | Salt stress | *Oryza sativa* | Chauhan et al., 2019 [44] |
| *Glutamicibacter* | Salt stress | *Oryza sativa* | Ji et al., 2020 [45] |
| *Pseudomonas* *Trichoderma (fungus)* | Drought | *Oryza sativa* | Singh et al., 2020 [46] |
| *Enterobacter* *Achromobacter* | Drought | *Oryza sativa* | Danish et al., 2019 [47] Danish et al., 2020 [48] |

**Table 1.** *Cont.*

| Bacterial Genera | Soil-Related Problem | Targeted Crops | Reference |
|---|---|---|---|
| *Bacillus*<br>*Enterobacter* | Drought | *Triticum aestivum*<br>*Zea mays* | Jochum et al., 2019 [49] |
| *Bacillus* | Drought | *Glycyrrhiza uralensis* | Zhang et al., 2019 [50] |
| *Agrobacterium Leclercia*<br>*Pseudomonas*<br>*Bacillus* | Drought | *Triticum aestivum* | Danish and<br>Zafar-ul-Hye, 2019 [51]<br>Zafar-ul-Hye et al., 2019 [52] |
| *Pseudomonas* | Heat | *Triticum* | Ashraf et al., 2019 [53] |
| *Bacillus* | Fungicidal toxicity and soil-borne diseases | *Oryza sativa* | Shen et al., 2019 [54] |
| *Bacillus* | Fungicidal toxicity and soil-borne diseases | *Arachis hypogaea* | Ahmad et al., 2019 [55] |
| *Nostoc*<br>*Anabaena* | Fungicidal toxicity and soil-borne diseases | *Oryza sativa* | Zhou et al., 2020 [56] |

## 3. Plant Growth-Promoting Rhizobacteria, A Potential Approach for Bioremediation

One of the safer tools to alleviate environmental deterioration is the application of ecofriendly living agents such as bacteria, fungi, algae and higher plants to eliminate the toxic chemicals, oil spills and toxic metals present in the altered sites, which is known as "bioremediation" [57]. To discriminate between the use of these biological agents for polluted sites' decontamination and their use in biorecycling processes designed to reduce the emission of toxins at source, bioremediation is defined as "a biological response to environmental abuse" [58,59].

Apart from their implications as plant growth enhancers, PGPR are also known to have a primordial role in practically all bioremediation aspects of contaminated soils (PTE, fungicides, organic pollutants, etc.) [60–62]. For example, some PGPR can interact with the soil's toxic metals by binding them to their cell surfaces or incorporating them in some metabolic functions after captivation. Some microorganisms can reduce metal ions such as $Hg^{2+}$ and $Ag^+$ to $Hg^0$ and $Ag^0$ and provide a perfect model for total metal removal from the soil [63]. It has been experimentally demonstrated that some rhizobacteria produce extracellular enzymes such as peroxidases, reductases, Cytochome $P_{450}$, lacases and glutathione-S-transferase, having important roles in polycyclic aromatic hydrocarbon degradation, molecules present in crude oil and known for their toxicity and carcinogenicity [64]. Meliani [65] produced a detailed review about PGPR application in soil decontamination, focusing on the genus *Pseudomonas*, and some of the most important strategies used by PGPR in bioremediation such as the production of biosurfactants, biofilms, toxic metal solubilization and siderophore production.

Plant Growth-Promoting Rhizobacteria utilize a wide range of mechanisms to improve plant growth and soil quality under stress conditions. Under ionic stress, mostly related to high salinity, drought and desiccation, certain bacteria can provide compatible solutes (proline, glycine betaine, sugars and derivatives, etc.) for root cells to avoid ion accumulation in the cytoplasm and, thus, water deficiency. Others can synthesize exopolysaccharides (EPS) in the rhizosphere. Bacterial EPS enhance water and ion ($K^+$, $Ca^+$, $Na^+$) uptake. They also play a major role in soil structure stabilization and aggregation under high ion concentrations. Certain PGPR express ion antiporters in their membranes ($Na^+$ ($K^+$)/$H^+$) to maintain their water balance in the cytoplasm under ionic stress. They also produce stress mitigation enzymes such as 1-aminocyclopropane-1-carboxylate (ACC) deaminase (see Section 5.2) and antioxidant enzymes such as peroxidase (POD), catalase (CAT) and superoxide dismutase (SOD) to eliminate the high amounts of reactive oxygen species (ROS) produced under abiotic stresses. In addition, PGPR phytohormones such as auxins, gibberellins, abscisic acid and cytokines (see Section 5.4) are inevitable for the stabilization of plants' physiology under water stresses [66]. In PTE bioremediation, several mechanisms are used by beneficial soil bacteria to adsorb, transform and uptake toxic elements in the

soil. In this context, the toxic species is either passively adsorbed (biosorption) to the cell surface or internalized (bioaccumulation) to its interior through "metabolic independent mechanisms". Inside the cytoplasm, toxic metal behavior follows more complex and, often, "metabolic dependent mechanisms" that may include: (1) metal compartmentation within specific organelles, (2) enzymatic detoxification (methylation, oxidation, dealkylation, reduction, etc.), and (3) efflux pumps that transport the modified and harmless toxic metal forms outside the cytoplasm (Figure 1) [67].

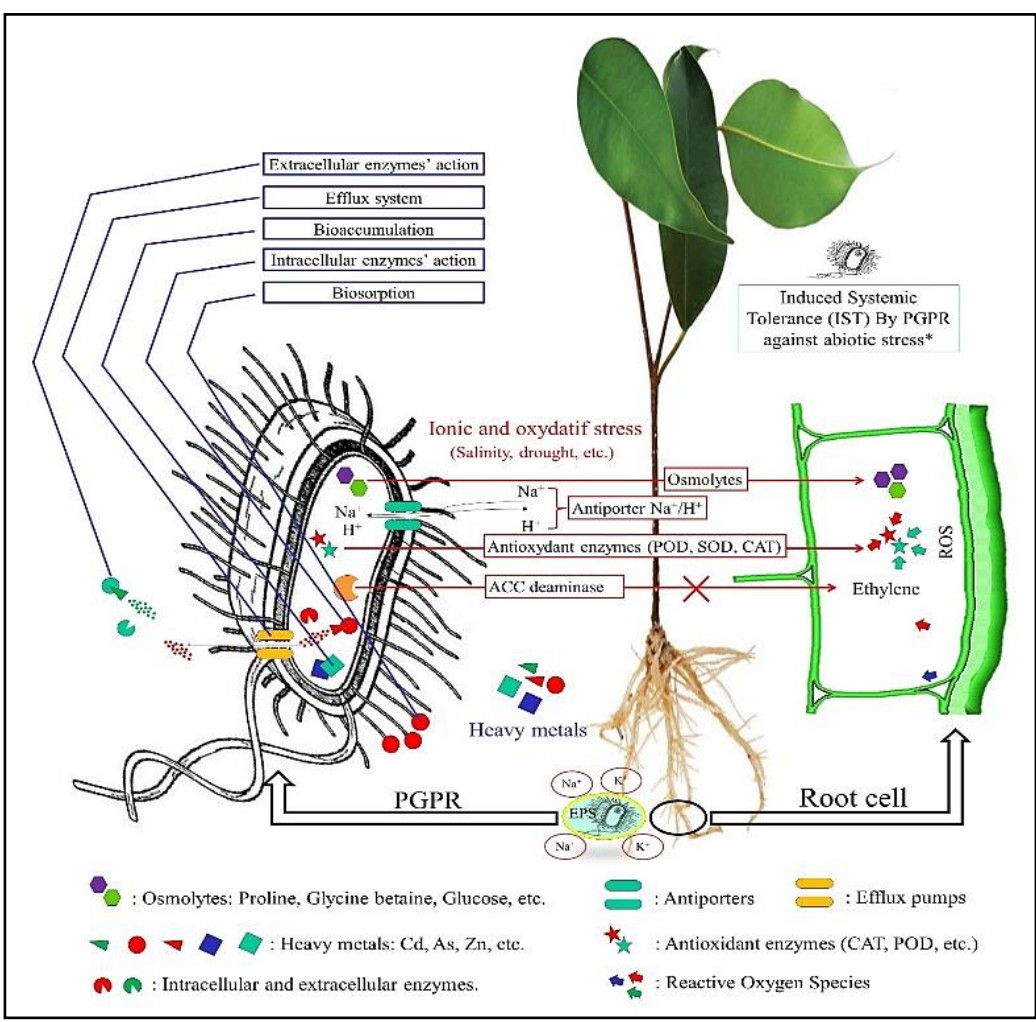

**Figure 1.** Principal mechanisms of bacterial bioremediation and plant growth regulation under abiotic stresses. * (IST): see Section 3.

At a cellular level, microbial populations permanently communicate between themselves, but at the same time with their surrounding root systems. In addition, some positive PGPR effects are controlled by small diffusible signaling molecules, mainly N-acyl homoserine lactone molecules (AHLs), that regulate gene expression in response to bacterial population density and interactions with plants, particularly under stress conditions [68,69]. For example, the N-3-oxo-hexanoyl homoserine lactone (3OC6-HSL) was found to be highly effective in restoring wheat root length, shoot length and fresh weight under salt stress conditions. In fact, 3OC6-HSL upregulated salt-responsive gene expressions such as (1) Abscisic Acid (ABA)-dependent osmotic stress genes: *COR15a, RD22, ADH* and *P5CS1*, (2) ABA-independent gene: *ERD1*, and (3) ion-homeostasis regulation genes: *SOS1, SOS2* and *SOS3* in Arabidopsis under salt stress conditions [70]. Moreover, Sheng et al. [71] carried out impressive work on the role of quorum sensing systems during microbial biofilm formation while degrading the pollutant 1,2,4-trichlorobenzene (1,2,4-TCB). For them,

there was no doubt about the positive correlation between the 1,2,4-TCB mineralization, microbial biofilm abundance and AHL (3-oxo-$C_{12:1}$-HSL; 3-oxo-$C_{10:1}$-HSL, 3-oxo-$C_{14:1}$-HSL; OH-$C_{14:1}$-HSL, etc.) production.

Among other PGPR, a *Pseudomonas aeruginosa* PS1 significantly promoted root and shoot nitrogen, root and shoot phosphorus and the seed yield of greengram (*Vigna radiata*) plants at all tested concentrations of tebuconazole, a fungicide belonging to the triazole group and largely used in agriculture. At high concentrations, tebuconazole may be accumulated in soils and plants, becoming toxic, damaging plant tissues and affecting crop yield [60]. Bensidhoum et al. [72] studied the effect of a *Pseudomonas protegens* S5LiBe on barley growth restoration under toxic metal contamination. The results showed an increase in germination rate, shoot and root fresh weight, shoot and root dry weight and shoot length. Moreover, the two siderophore producers, *Alcaligenes feacalis* RZS2 and *Pseudomonas aeruginosa* RZS3, showed a high ability to promote wheat growth when seeds were sown in PTE-contaminated soil; their bioremediation potential was higher than other metal chelators such as EDTA or citric acid [73]. Muratova et al. [74] realized a pot experiment in a growth chamber, proving the ability of *Azospirillum lipoferum* strain 5 to enhance the development of wheat root systems growing in crude oil-contaminated soil. In addition, a greenhouse experiment was conducted by Gomaa et al. [75] on the wheat growth stimulation potential of the two bacteria *Azospirillum lipoferum* and *Rhizobium leguminosarum* under different concentrations of Zn and Cd. The treatments: *Azospirillum*; *Azospirillum + Rhizobium*; *Rhizobium* + 200 ppm Zn; *Azospirillum + Rhizobium* + 300-ppm Cd; *Rhizobium* + 300 ppm Zn and *Azospirillum* + 300 ppm Zn resulted in an increase in wheat growth parameters compared to controls without bacterial inoculation. Similarly, a *Pseudomonas* sp. strain, isolated from hydrocarbon-contaminated soil, showed an important effect in stimulating rice's root and shoot elongation and enhanced its final yield in hydrocarbon and toxic metal-contaminated soils [76].

## 4. Plant Growth-Promoting Rhizobacteria Implication in Induced Systemic Tolerance and Induced Systemic Resistance

Currently, modern agriculture is facing extremely dangerous biotic and abiotic problems. Some of these constraints are the result of the human populations' growth explosion in some regions of the planet. Others are the direct consequence of environmental degradation due to drought, salinity, oxidative stress, toxic metals, nutrient deficiency and pathogens, loss of biodiversity and global climate changes [77–82]. In addition, the biggest part of the agricultural loss is due to abiotic factors. For example, the average yield of wheat is about 1500 kg/ha, while crop damage was estimated at 2000 and 14,500 kg/ha due to biotic and abiotic stresses, respectively [83].

Several works have highlighted the role of some PGPR as inducers of plant tolerance to abiotic stress by provoking physiological and biochemical changes in their tissues, which result in enhancing their tolerance to environmental stresses such as drought, salinity and toxic metals. Such complex interactions between plants and bacteria are known under the term "Induced Systemic Tolerance" (IST) [80,84–87]. Among other bacteria, *Arthrobacter*, *Azospirillum*, *Azotobacter*, *Klebsiella and Pseudomonas* are known for their ability to promote wheat and rice growth under high salinity conditions. [84,88–101].

Other works have described PGPR's effect on drought stress mitigation in wheat (*Burkholderia*, *Bacillus*, *Paenibacillus*, *Azospirillum* and *Azotobacter*) and rice (*Pseudomonas*, *Bacillus*, *Arthrobacter*, *Azospirillum*) [102–110]. Recently, PGPR conferring plant tolerance to toxic metals (Cd, Zn, etc.) have increasingly attracted researchers' attention. Thus, a *Pseudomonas* sp. SNA5 was efficiently used by Verma et al. [111] to promote wheat growth under high concentrations of cadmium, which is associated with high phosphate fertilization. Islam et al. [112] found that a *P. aeruginosa* was an ideal candidate for wheat growth enhancement against Zn-induced oxidative stress by improving the necessary nutrients' availability, as well as by lowering Zn metal uptake. In addition, ref [113] used PGPR to stimulate wheat growth under high Cr concentrations. Moreover, Gontia-Mishra et al. [114]

found that *Enterobacter ludwigii* (HG 2) and *Klebsiella pneumoniae* (HG 3) could significantly promote wheat seedling growth under mercury stress. Furthermore, the two bacteria *Ochrobactrum* sp. and *Bacillus* spp. revealed a high potential for rice growth promotion under toxic metal-contaminated soil [115].

It is largely recognized that many PGPR are implicated in plant defense stimulation against phytopathogens (viruses, bacteria, fungi and insects), which is designated "Induced Systemic Resistance" (ISR) [80,84–87]. Bacteria such as *Acinetobacter, Alcaligenes, Bacillus, Burkholderia, Enterobacter, Pantoea, Pseudomonas and Staphylococcus* have exhibited high antagonistic activities against *Alternaria alternata, Botrytis cinerea, Fusarium culmorum, F. oxysporum, F. solani, Gaeumanomyces graminis var. tritici, Phytophthora cryptogea, Pythium* and promoted wheat growth and defenses against these same phytopathogens [87,116–118]. Also, Rice inoculation with a combination of three PGP-*Pseudomonas fluorescens* (Pf1, TDK1 and PY15) increased chitinase accumulation and enhanced disease resistance in rice plants against sheath rot disease provoked by *Sarocladium oryzae* [119]. Elsewhere, PGPR such as *Bacillus, Pseudomonas, Rhizobium* and *Serratia* were used as biocontrol agents against rice pathogens (*Burkholderia glumae, Cnaphalocrocis medinalis, Cochliobolus myiabeanus, Hirschmanniella oryzae, Magnaporthe oryzae, Meloidogyne graminicola, Meloidogyne javanica, Pyricularia grisea, Rhizoctonia solani* and *Xanthomonas oryzae pv. oryzae*) [80,116,120,121].

## 5. Plant Growth-Promoting Metabolites for Soil Remediation and Crop Improvement

The ability of PGPR to fix nitrogen, solubilize nutrients and to produce other metabolites such as siderophores, phytohormones, antibiotics and hydrolytic enzymes makes them ecofriendly alternatives for avoiding the excessive use of unsuitable and cost-effective chemicals in agriculture (Figure 2).

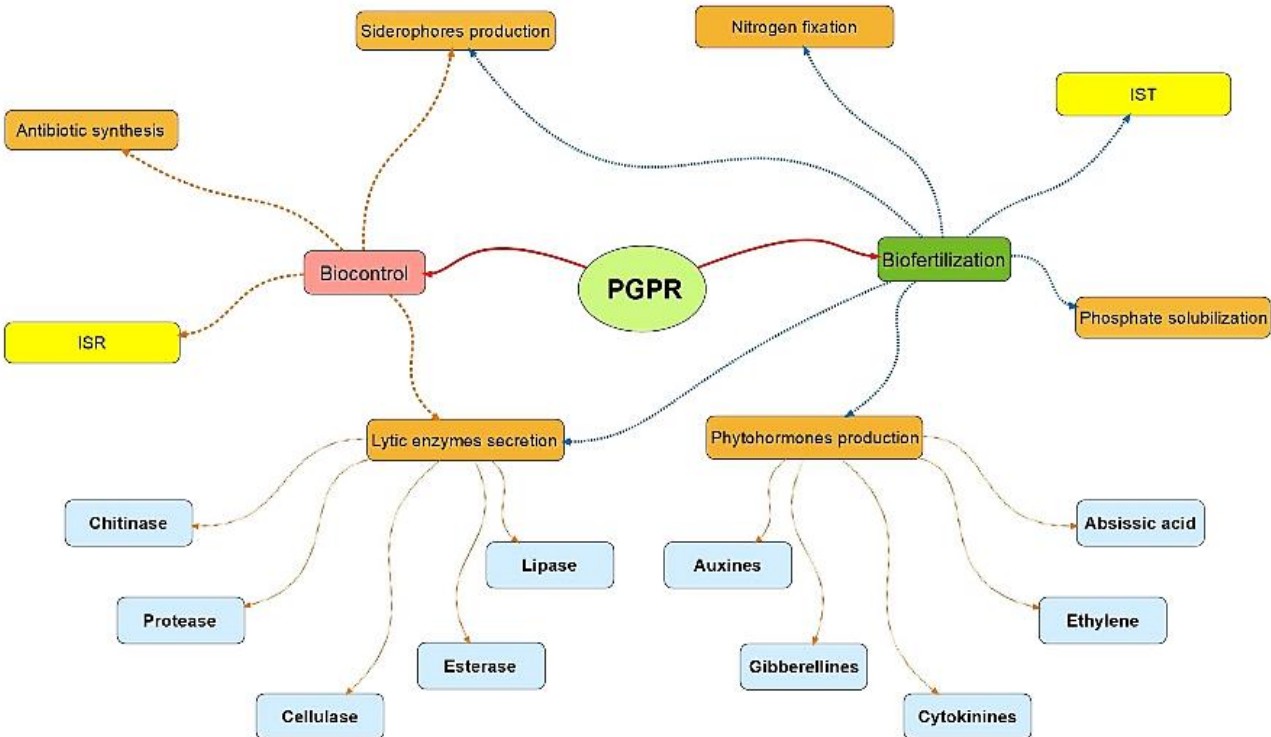

**Figure 2.** PGPR metabolites for soil remediation and crop improvement. ISR: Induced Systemic Resistance. IST: Induced Systemic Tolerance (realized using the Visual Understanding Environment software, VUE 3.3.0-developped by Tufts university, Medford, MA, USA).

### 5.1. Bacterial Siderophores in Soil

Iron is a key compound in almost all the electron transfer enzymes. In soil, $Fe^{3+}$ is the most found form (clays, oxides, hydroxides, etc.). The extremely insoluble nature of such forms constitutes a limiting factor for both plant and microorganism development [86]. Most soil bacteria and fungi, but also some plants, can produce chelators with a high affinity to iron. When iron is limited in soil, these chelators, namely siderophores, are secreted to collect it from various sources and deliver it to the producer, but also to some of the other cohabitating organisms [122]. Some siderophores producing PGPR can competitively participate in phytopathogen inhibition, especially when iron is limited in soil [123]. In addition, siderophores are also implicated in vegetal growth enhancement by providing accessible iron to plants [124]. Otherwise, some microorganisms produce siderophores with a high affinity to toxic metals, providing them with the ability to adsorb the metal in their biomass on metal-induced outer membrane protein or by bioprecipitation [125–127].

Two siderophore-producing and toxic metal-resistant bacteria, namely, *Enterobacter ludwigii* (HG 2) and *Klebsiella pneumoniae* (HG 3), were efficiently used by Gontia-Mishra et al. [114] to alleviate Hg toxicity in wheat (*Triticum durum*), suggesting their utility as potential candidates for Hg stress alleviation and wheat growth improvement. [115] studied the role of toxic metal-resistant and siderophore-producing *Ochrobactrum* sp. (CdSP9) and *Bacillus* spp. (AsSP9) strains in stimulating rice growth and remediating contaminated soils. The two strains reduced metal toxicity and enhanced overall rice biomass and root/shoot ratio.

In a work realized by Islam et al. [112], Zn-stressed wheat plants showed more green color intensity on their leaves and more iron availability when inoculated with a PGP-*Pseudomonas*, which was attributed to siderophore production by the bacteria. The bacterium also promoted wheat growth under greenhouse conditions, where shoot and root dry biomass was enhanced by 23% and 45%, respectively; they attributed this, in part, to its ability to produce high amounts of siderophores. Moreover, Rana et al. [128] attributed the significant role of a *Providencia* strain in the enhancement of wheat biomass, grain yield and macro- (NPK) and micronutrient contents to its ability to exhibit siderophore production, antifungal activity and synergistic interactions with other bacteria in wheat rhizosphere. A *Pseudomonas fluorescens* WCS374r was found to be implicated in the ISR of rice against *Magnaporthe oryzae*, which is based on pseudobactin (siderophore)-mediated priming for a salicylic acid-repressible multifaceted defense response [129]. Another *P. fluorescens*, together with its siderophore iron complex, was found to enhance phenol content and phenol-oxidizing enzyme content in rice plants, hence inducing systemic resistance in rice against the phytopathogen *Pyricularia oryzae* [130]. In addition, a complex of siderophore-producing PGPR consortia was used by Naureen et al. [131] to enhance rice growth and induce its systemic resistance to the phytopathogen fungus *Rhizoctonia solani*.

### 5.2. Plant Ethylene Balancing via Bacterial ACC Deaminase

Salinity affects approximately 6% of the land surface worldwide and about 20% of irrigated areas, posing a major threat to agriculture [15]. It affects plant growth and yield, but also provokes an imbalance of microorganism distribution in the rhizosphere and increases, in some cases, pathogen virulence [31]. Salinity also affects nutrient availability for crops. For example, phosphate (P) tends to precipitate with calcium in saline soils, which makes it difficult to be captured by plants [132], although some PGPR can be responsible for physical and chemical changes in plant tissues, participating in stress mitigation and plant growth restoration. The bacterial enzyme "1-amino cyclopropane-1-carboxylate (AAC) deaminase" plays an important role in regulating ethylene levels in plant tissues by degrading its precursor (ACC). Ethylene is a plant hormone that regulates several aspects of plant growth. However, its synthesis is accelerated under high salinity, and then it acts as a negative plant growth regulator as its concentration exceeds the required level [133].

The use of ACC deaminase-producing bacteria has been widely adopted in agriculture, especially for crop enhancement under different stresses [134]. Thus, Govindasamy et al. [135] reported the effect of ACC-deaminase-producing rhizobacteria on wheat growth promotion

under cadmium stress. The tested isolates significantly enhanced root elongation and minimized ethylene synthesis in wheat seedlings under such stressed conditions. Indeed, the most efficient strains, among others, in promoting wheat growth (*Pseudomonas* sp. PS 2–3 and *Pseudomonas fluorescens* PS 7–12) showed the presence of acdS gene coding for ACC deaminase. Abbas et al. [136] obtained better yield and chlorophyll content with rice inoculated with ACC deaminase-producing bacteria under salt stress. The ACC deaminase-producing *Pseudomonas putida* CEN7 and *Pseudomonas fluorescens* CEN8 showed a significant increase in the root length and root colonization of rice plants [137]. In addition, the bacterium *Rhizobium leguminosarum* (SN10) promoted the biomass, root branching and N content of four different rice varieties. Not only this, but the bacterium also displayed a strong chemotaxis response towards the rice seed and its root exudates [138]. Many other works have reported the role of ACC deaminase-producing PGPR in the enhancement of rice growth under different stresses such as salinity, flooding and toxic metals [137,139,140]. Moreover, similar results were found with wheat inoculated by ACC deaminase-producing bacteria such as *Pseudomonas*, *Serratia*, *Burkholderia* under salt and drought stress conditions [92,141–145].

*5.3. Phosphate Solubilization*

Phosphorus is one of the most important elements for plant nutrition. In agriculture, it is generally compensated through the addition of chemical fertilizers to soil. However, phosphorus coming from such fertilizers is rapidly immobilized, becoming useless for plants [146]. In addition, the high release of contaminants into the main product, gas steam and by-products, but also toxic metal accumulation in both the soil and crop due to the repetitive use of phosphoric fertilizers, has obliged producers to look for better tools to reduce the use of such chemical fertilizers [147,148]. Among these alternatives, the use of phosphate-solubilizing bacteria (PSB) is one of the most ecofriendly options to avoid or to minimize the exaggerated use of chemicals [149].

Ahemad [150] reviewed the role of metal phytoremediation in association with PSB and reported that such associations considerably overcome the practical drawbacks imposed by metal stress on plants. Furthermore, Paul and Sinha [151] isolated and selected a group of PSB with a high ability to tolerate toxic metal stress. They suggested that using toxic metal PSB in metal-contaminated areas might be exploited for bioremediation studies. Otherwise, Kaur and Reddy [152] found that inoculation with the two phosphate-solubilizing bacteria *Pantoea cypripedii* (PSB-3) and *Pseudomonas plecoglossicida* (PSB-5), together with rock phosphate fertilization, increased shoot height, shoot and root dry biomass, grain yield and total phosphorus uptake in both maize and wheat as compared to the control. The application of phosphate-solubilizing *Azotobacter* strains, alone or together with chemical fertilizers, improved the yield and root biomass of three wheat varieties under greenhouse conditions [153].

Several works have shown the efficiency of PSB such as *Pantoea*, *Azotobacter*, *Rhizobium*, *Pseudomonas* and *Serratia* in nitrogen uptake and wheat growth enhancement under different stress conditions [152,154–157]. The in vitro experiment, conducted by Panhwar et al. [158], to study the influence of two phosphate-solubilizing *Bacillus* spp. (PSB9 and PSB16), together with triple supper phosphate on aerobic rice growth, showed that the coupled "bacteria-triple supper phosphate" increased phosphate uptake, available soil phosphate and rice growth. The two phosphate-solubilizing bacteria PSB 12 identified as *Gluconacetobacter* sp. (MTCC 8368) and PSB 73 identified as *Burkholderia* sp. (MTCC 8369) were examined for their potential ability to enhance rice growth. They revealed high growth promotion ability under pot culture assays and were presumed to be of potential to develop as biofertilizers [159]. Vahed et al. [160] and Panhwar et al. [158] have also discussed the role of some PSB in improving phosphate uptake, soluble soil phosphate and rice growth.

### 5.4. Bacterial Phytohormones

Phytohormones are natural organic substances that influence plant development and regulate, at low concentrations, their physiology. The name auxin was given by Charles Darwin to the first discovered phytohormone, referring to "αυξειν", a Greek word that means grow or increase. Later, gibberellins, ethylene, cytokinin and abscisic acid joined auxins to be regarded as "the classical five phytohormones" [161,162]. Phytohormones affect several aspects of plant growth, nutrition, biotic and abiotic stress response and physiology. Their actions are in complex and continued interaction with each other, with plants, but also with the surrounding environment. Recently, phytohormones were classified, based on their physiological functions and structure, into different classes: abscisic acids (ABAs), auxins, cytokinins (CTKs), gibberellins (GAs), strigolactones, brassinosteroids, Jasmonic acid (JA), salicylic acid (SA) and ethylene [163]. While the hormonal classes are often associated with various characteristics and biological effects, increasing evidence suggests that multiple phytohormones often mediate the same biological processes by additive, synergistic or antagonistic actions, forming intricate signaling networks [164].

a.　　Auxins

Indole 3-Acetic Acid (IAA) is the most studied auxin. Its importance in promoting plant growth makes it an important line to select efficient PGPR [165–167]. Thus, the bacterial strains (*Pseudomonas putida, P. fluorescens* and *Azospirillum lipoferum*) were used by Sharma et al. [168] to enhance rice growth. The three bacteria showed a high ability to produce IAA and helped inoculated plants to express higher photosynthetic capacity and chlorophyll content, but also increased their root/shoot dry mass. Another *Pseudomonas putida* (BHUJY23) was found to produce high amounts of IAA and to be helpful for rice production and as an antagonistic agent against phytopathogens [169]. Hasan et al. [170] confirmed the beneficial effects of two IAA-producing bacteria, belonging to the genera *Rhizobium* and *Azospirillum*, on rice growth and yield. Bacteria belonging to the genera *Bacillus* and *Citrobacter* showed a significant improvement of root/shoot growth in inoculated rice plants. The results were attributed to IAA production by the studied isolates [171,172]. Torres-Rubio et al. [173], Tsavkelova et al. [174], Jha and Kumar [175], Soltani et al. [176] and Kumar et al. [177] reported that bacteria such as *Flavobacterium*, *Pseudomonas*, *Achromobacter* and *Azotobacter* can produce large quantities of IAA and promote wheat plant growth.

b.　　Gibberellins

The first gibberellin (gibberellic acid GA) was discovered in 1962 with the fungus *Fusarium moniliforme* (*Gibberella fujikuroi* in its sexual form), while the first report of bacterial gibberellins was in 1988 with the species *Rhizobium meliloti* [178,179]. Bacterial gibberellin synthesis starts with geranylgeranyl-PP conversion into ent-kaurene, which is then converted to GA12-aldehyde. After that, GA12-aldehyde is oxidized to $GA_{12}$ and metabolized to another GA [180]. Hasan et al. [170] studied the capacity of some PGPR strains (*Enterobacter* spp., *Azospirillum* spp.) to enhance rice growth under controlled conditions, alone and in combination with *Azospirillum or rhizobium*. Their results revealed a significant increase in gibberellic acid content in both the shoots and roots of the inoculated plants. Similar results have been reported by Caba et al. [181]. Moreover, Inoculation by the gibberellin-producing *Azospirillum* sp. and *bacillus* sp. resulted in increasing nitrogen uptake by wheat roots [182]. Furthermore, the water stress alleviation in wheat by PGPR was partially attributed to bacterial gibberellin production [183,184]. Many other reports have mentioned the beneficial effect of gibberellins on wheat and/or rice growth [185–190].

c.　　Cytokinins

Cytokinins are an important trait to search for in PGPR selection. They play a crucial role in the control of plant cell division, cell cycle, leaf senescence and nutrient mobilization, shoot apical meristem formation, seed dormancy and germination, floral development, etc. Chemically, cytokinins are N6-substituted aminopurines or adenine compounds with an

isoprene, modified isoprene, aromatic side chain attached to the N6-amino group or zeatin and trans-zeatin [86,179,191].

Bacteria such as *Azospirillum* [192], *Agrobacterium* [193], *Azotobacter* [194], *Pseudomonas* [193], *Paenibacillus* [195], *Achromobacter* [196], *Enterobacter* [197], *Bacillus* [198] and *Klebsiella* [192] are known for their implication in plant growth regulation via cytokinin production. Zahir et al. [199] conducted an interesting experiment to study the effect of kinetin (a synthetic cytokinin) and its physiological precursors (adenine + isopentyl alcohol) on rice growth. The results showed that the precursor was more effective than kinetin on yield improvement. It significantly enhanced plant height, tiller and panicle number, paddy yield and (NPK) content, hence suggesting that the precursor of kinetin was converted to cytokinin by rhizospheric microorganisms before absorption or transformed into cytokinins inside plant tissues. Otherwise, Kudoyarova et al. [200] studied the separate effect of the synthetic cytokinin trans-zeatin, *Bacillus subtilis* IB-22 (able to produce zeatin-type cytokinin) or *Bacillus subtilis* IB-21 (unable to produce cytokinins) on amino acid release from wheat roots in a split-root system. This system allowed the spatial separation of the zeatin or rhizobacterial application to one compartment and analyses of amino acid rhizodeposition into the other compartment. The application of *B. subtilis* IB-22 or trans-zeatin greatly enhanced amino acid liberation in the soil solution compared to *B. subtilis* IB-21 or the untreated plants, suggesting that the ability of cytokinin-producing *B. subtilis* IB-22 to enhance rhizodeposition may constitute an important process in improving the rhizobacterial colonization of the wheat rhizosphere.

d.    Abscisic acid

Abscisic acid (ABA) is the most important hormone produced by plants in response to abiotic stresses. However, ABA is also synthesized by bacteria, fungi, algae and animals. The prokaryotic pathway for synthesizing this fifteen-carbon sesquiterpene originates from isoprene, known as isopentenyl pyrophosphate, which is synthesized from the mevalonate pathway [201,202].

Travaglia et al. [203] investigated the influence of abscisic acid on wheat physiology and yield under field conditions with limited amounts of water during anthesis and postanthesis. Abscisic acid application significantly enhanced leaf area, chlorophyll and carotenoid content in flag leaf and soluble carbohydrates in shoots at anthesis. Yang et al. [204] found that the application of exogenous abscisic acid significantly promoted proline accumulation (as a response to water stress and/or dark-induced senescence) in the detached rice leaf when investigated in both dark and light conditions. Moreover, Azuma et al. [205] studied the promotive effect of abscisic acid on floating rice growing at a low relative humidity and its interactive role together with ethylene and gibberellin in internodal elongation. Their results revealed that the separate application of the abscisic acid had no promoting effect on the internodal elongation of the stem sections. However, the simultaneous application of abscisic acid, gibberellin and ethylene efficiently enhanced the internodal elongation of the stem sections, suggesting that abscisic acid could be a good enhancer of the ethylene- and gibberellin-induced internodal elongation at low humidity, hence preventing the reduction of water potential via apoplast transpiration.

*5.5. Bacterial Nitrogen Recycling for Soil Maintenance and Crop Improvement*

Nitrogen makes up 78% of the atmospheric volume and constitutes a major limiting factor for many physiological processes in soil. Molecular nitrogen cannot be assimilated by photosynthesizing plants or the majority of microorganisms [206]. The inter-conversion between nitrogen forms (nitrogen fixation, mineralization, nitrification and denitrification) represents the biogeochemical cycle of nitrogen, mainly made of biological processes in which microorganisms play a predominant role. Metabolically, microorganisms preferentially uptake ammonium while plants and some microorganisms assimilate nitrate [207].

Nitrogen fixation is the first step in the process of making nitrogen usable by plants, where nitrogen-fixing bacteria play a crucial role in changing dinitrogen into ammonium via nitrogenase activity. After that, the produced ammonium needs to be converted into

nitrates before assimilation by plants. This nitrification process, in which bacteria play an important role, allows nitrogen assimilation by plant roots to be used in amino acids, nucleic acids and chlorophyll composition. When a plant or animal dies, decomposers such as fungi and bacteria turn nitrogen back into ammonium so it can reenter the nitrogen cycle through denitrification [208].

Unfortunately, the excessive use of fertilizers to supply nitrogen in soil and nitrous oxide emission in the atmosphere due to other human activities results in the unbalancing of the nitrogen cycle in both the atmosphere and soil [209]. To avoid such exaggerated application of chemical nitrogen to soil, N-fixing bacteria such as *Azospirillum* [210–215], *Pseudomonas* [210,211,216,217], *Bacillus* [218], *Herbaspirillum* [215,219], *Azotobacter* [210,220], *Rhizobium* and *Enterobacter* [221], *Klebsiella* [222] and *Burkholderia* [223,224] have proved their efficiency in reducing or replacing chemical fertilizers for wheat and rice crop enhancement.

In addition to nitrogen fixation, the presence of a positive correlation between bacterial denitrification and the rhizosphere colonization potential has permitted researchers to consider denitrification as an important trait in isolating and selecting efficient PGPR. In this context, a fluorescent PGP-*pseudomonas* associated with rice was isolated and selected by Kumar et al. [177], considering the denitrification potential as an important trait in selecting competitive PGPR. Moreover, Muriel et al. [225], considered denitrification in the PGP-*Pseudomonas fluorescens* F113 as an important character. The NO produced by the PGP-*Azospirillum brasilense* Sp245 via denitrification could be a major signal implicated in wheat root branching stimulation when the dissimilatory nitrite reductase gene (nirK) is upregulated [226].

## 6. Conclusions

In the last few decades, soil salinization and contamination with petroleum, hydro-carbons and toxic metals have seemed to be directly linked to some environmentally uncontrolled anthropogenic activities and population expansion. In arid environments, such alarming problems have become a major threat to global cereal production. In addition, applying physicochemical techniques to maintain soil health and ensure enough food production is often disruptive, labor intensive and relatively expensive. Recently, PGPR application for soil bioremediation and cereal growth improvement has received considerable attention for its ecofriendly, efficient and cost-effective advantages. Thus, bacteria such as *Arthrobacter, Azotobacter, Bacillus, Enterobacter, Pseudomonas,* etc., have proved their efficiency as plant growth promoters and soil quality remediators in arid environments. However, most of the reported experiments have been realized in lab- or greenhouse-controlled conditions, and we still lack information about their interactions with plants and other microorganisms once in the field, certainly a more complex environment. Progressively, and with increasing knowledge about the interactive aspects between plants, bacteria and soil, but also the understanding of the key signal molecules implicated in such interactions, PGPR's place in modern agriculture is now undeniable as biocontrol, biofertilization and bioremediation agents. However, more investigations are needed for a better understanding of some problems related to bacterial long-term stability and efficiency in the field, their large-scale production once selected, their storage, transportation and delivery conditions, but also their long-term effects on the innoculated environment. Moreover, incorporating more data about the impact of aridity expansion on soil composition, and both microbial and plant diversity, is crucial to direct the already obtained results for better PGPR applications.

**Author Contributions:** All authors have contributed substantially to the work. All authors have read and agreed to the published version of the manuscript.

**Funding:** This research received no external funding.

**Institutional Review Board Statement:** This work is a literature review that does not require ethical approval. The submitted work is a review. It does not contain any human/animal experiments. The manuscript is rriginal; it is not submitted to another journal for simultaneous consideration and is not published elsewhere in any form or language.

**Informed Consent Statement:** This work is a literature review that does not require any consent statement.

**Data Availability Statement:** Not applicable.

**Conflicts of Interest:** The authors declare no conflict of interest.

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
