# Peer review of "Bacteria in Soil: Promising Bioremediation Agents in Arid and Semi-Arid Environments for Cereal Growth Enhancement"

_applsci, doi:10.3390/app122211567_

Round 1

Reviewer 1 Report

The manuscript titled Bacteria in soil: promising bioremediation-agents in arid and semi-arid environments for cereal growth enhancement required minor revision. 

This work reviewed understanding the prospective functions of Plant Growth Promoting Rhizobacteria (PGPR) and which can participate in soil bioremediation, stress alleviation and plant growth restoration.

It addressed a specific gap in the field. The authors have tried to highlight the importance of using beneficial soil bacteria for both soil quality restoration and plant growth enhancement which has not been investigated in arid environments.

Further control isn't necessary for this study. The authors need to describe the search strategy.  The review has no protocol in methodology showing which databases have been used to search. I would suggest authors include a search string with more details. A flowchart would be useful.

Also, it is required to provide alphabetical order abbreviations. In addition, please remove the extra “in” line 484. Conclusions consistent with the evidence presented; references appropriate; The tables and figures make sense. Line 104-105 needs to be modified. Line 103, please replace participat with participate.

Reviewer 2 Report

This is an interesting and good topic. Author have done a good review about it. Still, some need to be revised.

1. The format needs to be revised and modified throughout the whole manuscript, like the Line 17, Line 34,35 with ";" not ","

     any Correspondence? for there are no information about this. 

    Spaces are required at the beginning of each paragraph.

    the citation method seems wrong.

2. The conclusion needs re-write. it's not well.

3. I glanced the references, seems less recently 2 years papers. As a reviewer, this also need add in.

4. Some expression need revised.

5. These should be alphabetical order, like L30, L237, L262,263, L271, L570 etc.

6. The expression of L220 "PGPR between induced systemic tolerance and resistance " seems got problem.

7. L522 "1.1. Bacterial nitogen recycling for soil maintenance and crop improvement" ?why 1.1?

8. Fig 1,2 should be "Figure 1". all like this.

9. The abstract and conclusion are not well written.

10. Though the authors have metioned many exsit effect from PGPR, but the expression can be more deeper. As a review, this should be well refined.

11. All the references' fomate are totally wrong.

     In fact, author should pay more attention to orgnize it.
